# What are the experiences of seeking, receiving and providing FGM-related healthcare? Perspectives of health professionals and women/girls who have undergone FGM: protocol for a systematic review of qualitative evidence

Catrin Evans,[1] Ritah Tweheyo,[1] Julie McGarry,[1] Jeanette Eldridge,[2] Carol McCormick,[3] Valentine Nkoyo,[4] Gina Marie Awoko Higginbottom[1]

For numbered affiliations see end of article.

**Correspondence to**
Dr Catrin Evans;
catrin.evans@nottingham.ac.uk

## ABSTRACT

**Introduction** Female genital mutilation (FGM) is an issue of global concern. High levels of migration mean that healthcare systems in higher-income western countries are increasingly being challenged to respond to the care needs of affected communities. Research has identified significant challenges in the provision of, and access to, FGM-related healthcare. There is a lack of confidence and competence among health professionals in providing appropriate care, suggesting an urgent need for evidence-based service development in this area. This study will involve two systematic reviews of qualitative evidence to explore the experiences, needs, barriers and facilitators to seeking and providing FGM-related healthcare in high-income (Organisation for Economic Cooperation and Development) countries, from the perspectives of: (1) women and girls who have undergone FGM and (2) health professionals.

**Review methods** Twelve databases including MEDLINE, EMBASE, PsycINFO, ASSIA, Web of Science, ERIC, CINAHL, and POPLINE will be searched with no limits on publication year. Relevant grey literature will be identified from digital sources and professional networks. Two reviewers will independently screen, select and critically appraise the studies. Study quality will be assessed using the Joanna Briggs Institute Qualitative Assessment and Review Instrument appraisal tool. Findings will be extracted into NVivo software. Synthesis will involve inductive thematic analysis, including in-depth reading, line by line coding of the findings, development of descriptive themes and re-coding to higher level analytical themes. Confidence in the review findings will be assessed using the CERQual approach. Findings will be integrated into a comprehensive set of recommendations for research, policy and practice.

**Dissemination** The syntheses will be reported as per the Enhancing Transparency in Reporting the Synthesis of Qualitative Research (ENTREQ) statement. Two reviews will be published in peer-reviewed journals and an integrated report disseminated at stakeholder engagement events.

### Strengths and limitations of this study

► Will illuminate health professional and organisational factors that influence choices and behaviour on providing female genital mutilation (FGM) care and following FGM management protocols in a range of healthcare settings (not just maternity settings).

► Will develop an in-depth understanding of women's/girl's care seeking choices, barriers and experiences across the life course (not just maternity settings).

► Will synthesise research in similar high-income contexts so that findings can be directly translated into interventions and service initiatives across Organisation for Economic Cooperation and Development countries that share similar challenges in service provision and uptake.

► This is a participatory collaborative project that involves community representatives from identification of the initial questions through to dissemination.

► The systematic reviews will only include qualitative empirical evidence. Hence, a limitation is that opinion pieces, editorials and case studies of personal or professional experiences will be excluded.

**PROSPEROregistration number** CRD42015030001: 2015 and CRD42015030004: 2015.

## INTRODUCTION

Female genital mutilation (FGM) refers to all procedures that involve the partial or total removal of the external female genitalia or other injury to the female genital organs for non-medical reasons.[1] The practice is illegal in most countries and is an internationally recognised violation of the rights of women

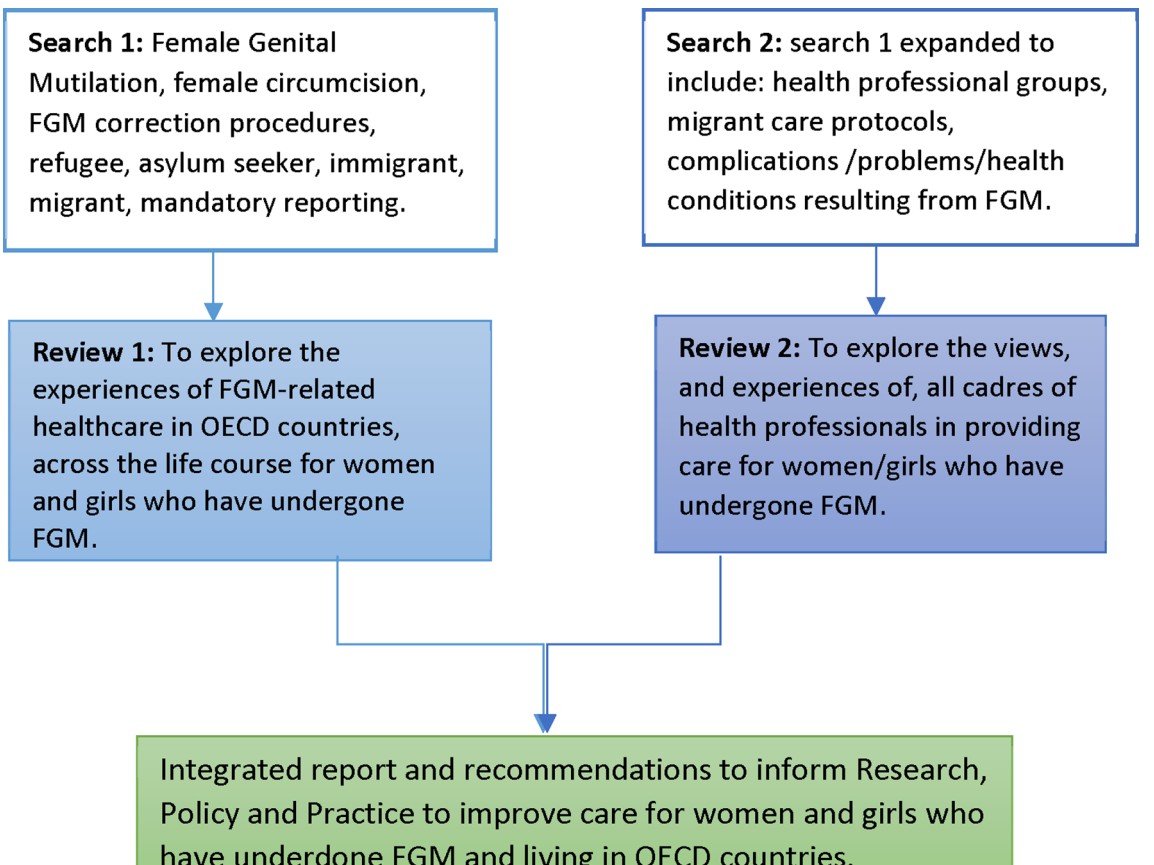

**Search 1:** Female Genital Mutilation, female circumcision, FGM correction procedures, refugee, asylum seeker, immigrant, migrant, mandatory reporting.

**Search 2:** search 1 expanded to include: health professional groups, migrant care protocols, complications /problems/health conditions resulting from FGM.

**Review 1:** To explore the experiences of FGM-related healthcare in OECD countries, across the life course for women and girls who have undergone FGM.

**Review 2:** To explore the views, and experiences of, all cadres of health professionals in providing care for women/girls who have undergone FGM.

Integrated report and recommendations to inform Research, Policy and Practice to improve care for women and girls who have underdone FGM and living in OECD countries.

**Figure 1** Systematic reviews to improve FGM-related care. FGM, female genital mutilation; OECD, Organisation for Economic Cooperation and Development

and girls.[2] FGM is associated with significant negative physical, psychological and sexual health sequelae.[3] In the immediate and short term, these can include shock, infection, urinary retention or injury to other tissues (eg, vaginal fistulae).[4 5] In the longer term, they can include psychological problems, post-traumatic stress disorder, painful intercourse and other sexual problems, relationship problems, chronic pain, chronic infections, infertility and complications in childbirth.[1 5] It is essential therefore that affected women and girls have access to high-quality services that can identify and meet these multiple complex health needs and that include mental as well as physical healthcare provision.[1 6 7]

There are 30 countries where FGM is traditionally practised, with over 200 million women and girls affected worldwide, mainly from Africa and parts of Asia.[2] However, increased migration means that FGM is now regularly seen within health services in higher income 'receiving' countries where refugees and migrants from practising countries have settled.[2 8] It has been estimated that over half a million women and girls residing in European Union (EU) countries are FGM survivors.[8] Within the UK, it is thought that approximately 137 000 women and girls living in England and Wales have undergone FGM,[9] and that, since 2008, women with FGM make up approximately 1.5% (nearly 11 000) of all maternity episodes.[9] Hence, health services and health professionals

in receiving countries are increasingly being challenged to develop appropriate services and to care for women/ girls who have undergone FGM in a culturally sensitive manner.[6 10] Migrant's access to healthcare in these countries can be problematic, however, due to a range of socioeconomic, legal, language and cultural barriers.[11 12] The ways in which these factors intersect to affect appropriate care delivery for women/girls who have undergone FGM—a particularly sensitive and personal issue—is currently unclear.

Existing evidence on women's and health professionals' experiences around FGM is primarily orientated to maternity care delivery, with much less attention devoted to care delivery in other settings, or for other problems or at other stages of the life course.[1 13] The available evidence from women, however, highlights a number of concerns, including not knowing about specialist services, feeling unable to talk about FGM, feeling judged and experiencing fear, pain and isolation.[14–22] Likewise, evidence from health professionals indicates that they lack confidence and competence in caring for, and talking about, FGM.[10 23–26] Interestingly, even where training has been given and where clear care pathways and protocols exist, problems may still remain. For example, a recent study of FGM management in a large London maternity unit found that, in spite of the existence of guidelines and training, clinical care for women/girls with FGM was

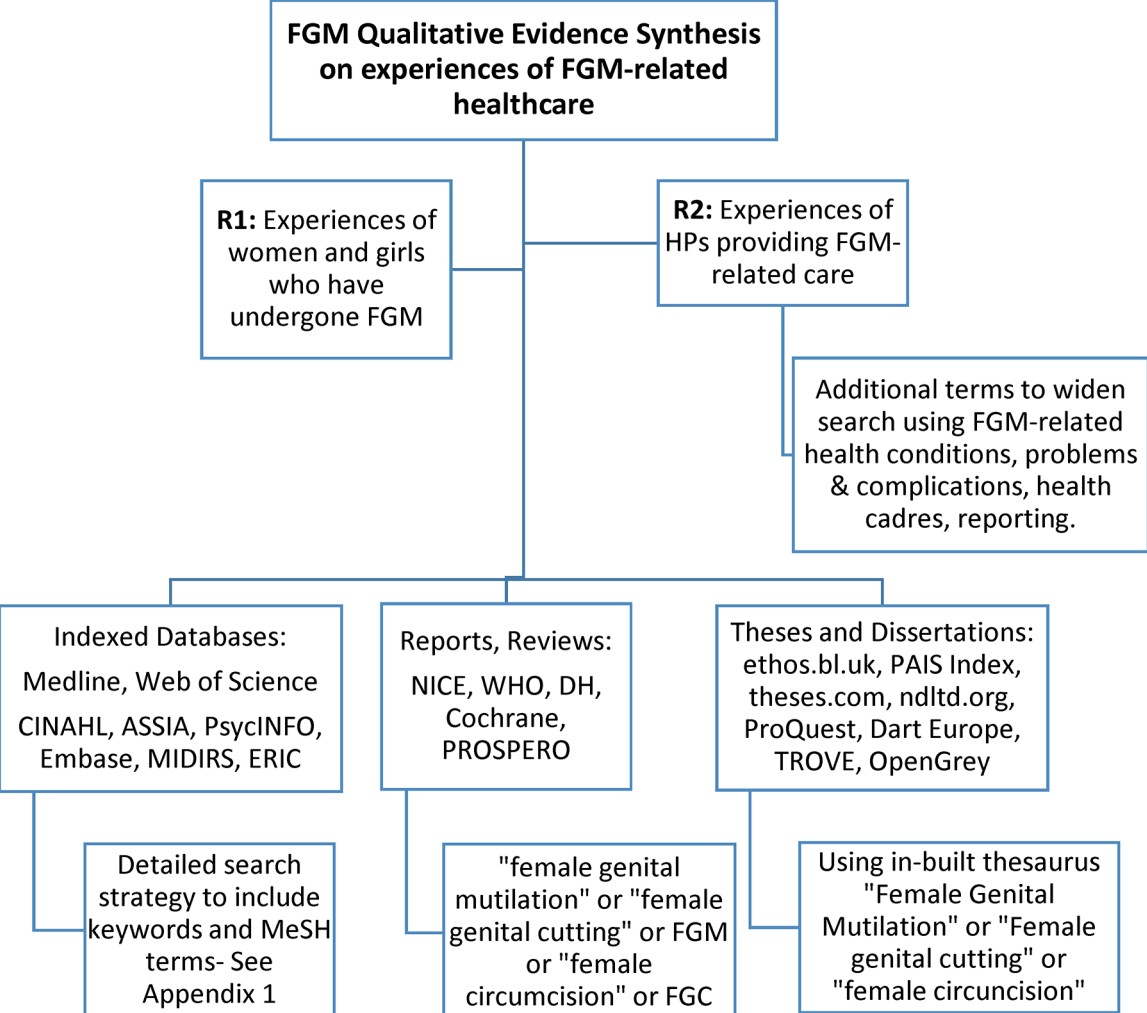

**Figure 2**  Search summary flowchart.

suboptimal.[27] The maternity unit had access to a FGM specialist service, but 41% of women with FGM were not identified until they arrived in the labour ward. Hence, even though a specialist service existed, it was not being optimally used to benefit women with FGM, and a significant percentage of opportunities were missed to provide women with specialist care. Similar findings were reported from a study in a maternity unit in Switzerland where, in spite of staff training and the existence of clear guidelines, FGM was correctly identified and managed in only 34 (26.4%) of 129 cases reviewed.[28] The reasons for this lack of adherence to protocols are unclear, however, and there is a need to explore in more depth how organisational and personal factors may influence health professionals' views and behaviour in this area.[10 24 25 29–36]

In the UK, a range of policies and protocols around FGM prevention and care have been developed over the last decade. These set out safeguarding procedures to mitigate risk of FGM and recommend a multiagency approach to service development.[6 37] In the UK, these include a declaration to end FGM in 2014, the establishment of a National FGM Prevention Programme, development of Intercollegiate Guidelines,[38–40] mandatory

recording using the SCCI2026 FGM Enhanced Dataset Information Standard, the Serious Crime Act 2015 and mandatory reporting of girls at risk of FGM.[41] Many other Organisation for Economic Cooperation and Development (OECD) countries have similar laws barring FGM, including direct prosecution of individuals continuing the practice.[37 42] All EU countries consider FGM a crime of serious bodily harm, with legal measures to safeguard minors and professional provisions for disclosure and duty to report.[37] Approaches to prosecution are varied, resulting in calls for efforts to improve and correct the inadequate care of women and girls who have undergone FGM,[43] specifically for multidisciplinary approaches.[13 44] It is currently unknown how these legal and regulatory provisions impact on the experience of seeking and providing healthcare from patients' and health professionals' perspectives.

Much of the existing body of research around FGM relates to understanding the practice of FGM,[45] the prevention of FGM[46 47] and the psychosocial and clinical consequences of FGM.[3 5 48 49] Evidence on the effectiveness of FGM-specific interventions or models of care is currently lacking,[50 51] as concluded by a

| Screening Criteria | Yes | No | Unsure |
|---|---|---|---|
| Does the study report findings related to………………<br>• Experiences of seeking and receiving **healthcare** related to FGM?<br>• Experiences of providing **healthcare** to women/girls who have undergone FGM? | | | |
| Is the study conducted in an OECD country? | | | |
| Is it a qualitative study?<br>Or<br>Does it report qualitative findings from a mixed methods study? | | | |
| Are the participants…………………<br>• Women or girls who have undergone FGM?<br>• Health professionals? | | | |

**Figure 3** Screening checklist. FGM, female genital mutilation; OECD, Organisation for Economic Cooperation and Development.

series of 10 systematic reviews published by WHO in a recent journal supplement.[52] In the absence of high-quality evidence on the clinical or cost-effectiveness of different types of service provision, decision-making around FGM services must be informed by other forms of evidence.[53] Hence, this project proposes to bring together existing qualitative evidence on lay and professional experiences of, and perspectives on, FGM care and to relate this to service development in higher income (receiving) countries. In doing so, it will provide a vital resource for evidence-informed FGM-related service development and training. Qualitative research can provide insights into factors influencing service acceptability, appropriateness and meaningfulness and on micro level and macro level organisational and contextual issues that influence service use and delivery.[54–56] Therefore, the proposed qualitative evidence syntheses around FGM will generate rich data and deep understanding with which to inform new service initiatives and the content and structure of staff training resources.[57 58] The above-mentioned WHO systematic review series[50] included three qualitative reviews, but these were focused on evidence related only to specific interventions, rather than healthcare experiences more generally and hence identified a very limited number of studies.[59–61]

There are a number of systematic reviews related to FGM healthcare that have been published to date, but all have taken a multicontext (or 'lumping') approach to the evidence,[13 50 62–64] by including research from high-income and low-income settings across the world. Many key themes from these reviews therefore are drawn from evidence from very different contexts (see Sunday-Adeoye and Serour[52]) and are not easily transferable to a high-income setting. These countries generally have strong, well-resourced public health systems and FGM is primarily found within their migrant populations.[2] FGM care and its challenges in these settings, will, therefore,

be linked to other challenges around providing care for migrant populations (such as lack of familiarity with cultural norms, systems or communication issues[11]). This is a very different situation compared with countries where health systems are weak, where FGM is more prevalent, where health providers may have greater exposure to, and indeed, may even be complicit in, FGM (ie, medicalisation of FGM).[65] In order to address this shortcoming, the proposed systematic reviews of women's and health professionals' experiences will focus exclusively on evidence from high-income OECD contexts.

## AIM

The aim of this review will be twofold: to understand the experiences, needs, barriers and facilitators around seeking and providing FGM-related care in OECD countries from the perspectives of (1) women and girls who have undergone FGM and (2) health professionals. Hence, this will involve two reviews of qualitative evidence.

**Review 1**: *to explore the experiences of FGM-related healthcare in OECD countries across the life course for women and girls who have undergone FGM.*

### Objectives

From the perspective of women and girls who have undergone FGM

1. To illuminate factors that influence FGM-related healthcare seeking and access to health services across the life course.
2. To explore how quality of care is perceived and experienced in different healthcare settings and with different groups of healthcare professionals.
3. To characterise and explain elements of service provision considered important for the provision of acceptable and appropriate healthcare.
4. To describe factors perceived to influence open discussion and communication around FGM (including prevention) with health professionals.

| | Summary JBI Qualitative Assessment and Review Instrument (QARI) | | | | | | | | | | | | |
|---|---|---|---|---|---|---|---|---|---|---|---|---|---|
| Record details / Full reference | Is there congruity between the stated philosophical perspective and the research methodology? | Is there congruity between the research methodology and the research question or objectives? | Is there congruity between the research methodology and the methods used to collect data? | Is there congruity between the research methodology and the representation and analysis of data? | Is there congruity between the research methodology and the interpretation of results? | Is there a statement locating the researcher culturally or theoretically? | Is the influence of the researcher on the research, and vice-versa, addressed? | Are participants, and their voices, adequately represented? | Is the research ethical according to current criteria or, is there evidence of ethical approval by an appropriate body? | Do the conclusions drawn in the research report flow from the analysis, or interpretation, of the data? | Score | Seek further info | Comments (Including reason for exclusion) |
| | Q1 | Q2 | Q3 | Q4 | Q5 | Q6 | Q7 | Q8 | Q9 | Q10 | /10 | | |
| | | | | | | | | | | | | | |
| | | | | | | | | | | | | | |
| | | | | | | | | | | | | | |
| | | | | | | | | | | | | | |
| | | | | | | | | | | | | | |
| | | | | | | | | | | | | | |

**Figure 4** Extracted Joanna Briggs Institute quality appraisal tool.

**Review 2**: *to explore the views on, and experiences of, all cadres of health professionals in providing care across the life course in OECD countries for women/girls who have undergone FGM.*

**Objectives**

From the perspective of health professionals

i. To explore how quality of care for women/girls who have undergone FGM is perceived in different healthcare settings and among different professional groups.

ii. To characterise and explain elements of service provision considered important for the provision of high quality care to women/girls who have undergone FGM.

iii. To illuminate factors perceived to facilitate or hinder appropriate provision of care for women and girls who have undergone FGM.

iv. To identify processes and practices perceived to influence open discussion and communication around FGM (including prevention) with women/girls from affected communities.

## METHODOLOGY AND METHODS

The review methodology uses a participatory approach in which its aims and objectives were generated together with community organisations working in the field of FGM who continue to be involved as co-investigators and advisors.[66] The project originated from a request by a community organisation run by, and working with, FGM-affected women to explore women's and health professionals' experiences in the healthcare encounter related to FGM. The review team and advisory group includes members of FGM-affected communities and their perspectives will be integrated into project activities at every stage, for example, in identifying relevant grey literature, contributing to interpretation of key themes, helping to formulate recommendations and helping to access professional and community networks to aid dissemination and community engagement.

The reviews seek to identify insights about lay/health professional experiences of FGM-related healthcare and perceived appropriateness and acceptability of services. These are questions best answered by qualitative research[54 55]—hence the specific focus on qualitative evidence. There are many possible approaches to qualitative evidence synthesis, with most discussions in this area characterising the different types along a continuum between aggregation and interpretation.[67] Where the purpose of a synthesis is to generate new theoretical insights, a highly interpretive approach such as meta-ethnography may be most suitable, informed by an idealist epistemological stance.[68 69] However, where the purpose is to inform policy or practise a more aggregative or thematic approach informed by a realist epistemology is often advocated.[69] The latter is also suggested in cases where the existing evidence is likely to be descriptive (as in much health services research) rather than highly theoretical or conceptual.[57] An initial scoping of the literature suggests that this is the case for the proposed syntheses. A thematic synthesis approach involves using thematic analysis techniques to identify key concepts/themes within primary research studies.[70 71] Synthesis involves an iterative and inductive process of grouping themes into overarching categories and exploring the similarities, differences and relationships between them.[57 58] Thematic synthesis explicitly aims to move beyond generating a list of descriptive themes (as would be the case in meta-aggregation) in order to identify new, higher order, analytical insights that can contribute to new understandings of a phenomenon.[57] Review recommendations, however, are clearly formulated to inform policy and practice and to identify research gaps. As such, it is considered the most suitable approach for the two proposed systematic reviews.[72]

The findings of the two reviews will be reported separately and then, where appropriate, integrated (figure 1) to enable greater understanding of key issues or concepts when presented from multiple perspectives. The findings will be brought together into a comprehensive set of recommendations for service development, community

| Modified JBI data extraction form for FGM Systematic Review | | | | | | | | | | | |
|---|---|---|---|---|---|---|---|---|---|---|---|
| Study (Name and Authors) | Purpose / Phenomena of interest | Methodology (type, theory / framework) | Methods (type, recruitment and sampling) | Setting | Geographical (Country and Region) | Cultural | Participants (Age, relevant number, sample) | Data analysis | Findings/ Results | Authors Conclusion | Reviewers Comments |
| | | | | | | | | | | | |
| | | | | | | | | | | | |
| | | | | | | | | | | | |
| | | | | | | | | | | | |
| | | | | | | | | | | | |
| | | | | | | | | | | | |
| | | | | | | | | | | | |
| | | | | | | | | | | | |
| | | | | | | | | | | | |

**Figure 5** Data extraction tool to include all the results and findings sections of each included study.

engagement, health professional education and future research.

## Approach to searching

The project will implement a comprehensive search strategy to gather all available and accessible studies, including peer-reviewed articles and grey literature. Primary research articles in the form of journal papers, research/evaluation reports, theses and dissertations will be collected. Reference lists from primary papers and key reviews will be hand searched to identify additional papers. There will be no language or date limits.

## Data sources

Searching will be done in at least 12 databases to include Medline (Ovid), Embase, Scopus, PsycINFO, Web of Science, ASSIA, CINAHL, ERIC, MIDIRS using a detailed search strategy (see Appendix 1 in the online supplementary file 1). Additional searches will be carried out in relevant indexes such as POPLINE, grey literature databases including: British library, National Institute for Health and Care Excellence evidence services, index to theses, Networked Digital Library of Theses and Dissertations, ProQuest and other accessible digital thesis and dissertation repositories.

## Search strategy

The search strategy was developed from an initial scoping search undertaken to establish relevant search terms and potential databases and to develop a robust integrated, but specific and sensitive strategy. It is vital that we have a comprehensive search strategy because of known poor indexing of qualitative studies. Therefore, the scoping search will identify records which will be cross-checked for indexing, together with FGM reports and guidelines from WHO to ensure correct terms and FGM-related terms are captured. The Librarian (JE) together with RT will develop and conduct all the searches. The comprehensive search will be tailored to each of the listed databases for both peer reviewed and grey literature and results reported in a flowchart following Preferred Reporting Items for Systematic Review and Meta-Analysis Protocols guidance.

No limits will be placed on language or publication year in the search strategy. We will include articles available

from the start of database indexing to present (up to 31 December 2017). The scoping search will run from April to June 2017, and the expanded search will run from July to October 2017. We will set-up automatic search alerts and updates until 31 December 2017. Electronic searches are summarised in figure 2 and an initial detailed search strategy is attached in Appendix 1 in the online supplementary file 1.

## Screening and eligibility

All returned results from the searches will be entered into the EndNote reference manager programme. Screening will be undertaken independently by two reviewers using title and abstract and studies selected if they meet the a priori selection criteria (see figure 3). In case of disagreement, there will be recourse to a third reviewer when required.

The inclusion criteria are:
- ▶ Population: Women and girls who have undergone FGM, and their experience of FGM-related healthcare, or healthcare professionals or students involved in the care of women and girls that have undergone FGM.
- ▶ Phenomenon of interest: Experiences of seeking, receiving or providing FGM-related healthcare across the life course.
- ▶ Context or setting: Studies conducted in OECD/high-income countries as described by the World Bank (list attached in Appendix 1 in the online supplementary file 1).
- ▶ Study type: Any type of qualitative study and any type of mixed methods study that reports qualitative findings.
- ▶ Search limits: No language or date limits. The abstracts of articles not in English will be translated and assessed. Those that meet the screening criteria will then be professionally translated into English for appraisal and data extraction.
- ▶ Exclusion: Any papers or research that was not undertaken on women and girls who have undergone FGM, or healthcare professionals, or not in OECD countries. Studies will be excluded if they do not describe FGM-related experiences of healthcare or provision of healthcare.

**Table 1** Summary of the 21 items from the Enhancing Transparency in Reporting the Synthesis of Qualitative Research statement

| No. | Item |
|---|---|
| 1 | Aim |
| 2 | Synthesis methodology |
| 3 | Approach to searching |
| 4 | Inclusion criteria |
| 5 | Data sources |
| 6 | Electronic search strategy |
| 7 | Study screening methods |
| 8 | Study characteristics |
| 9 | Study selection results |
| 10 | Rationale for appraisal |
| 11 | Appraisal items |
| 12 | Appraisal process |
| 13 | Appraisal results |
| 14 | Data extraction |
| 15 | Software |
| 16 | Number of reviewers |
| 17 | Coding |
| 18 | Study comparison |
| 19 | Derivation of themes |
| 20 | Quotations |
| 21 | Synthesis output |

## Quality appraisal of studies

The role of critical appraisal in qualitative evidence synthesis is contested and there is lack of agreement over: (1) the appropriateness of excluding studies, (2) the potential impact (or not) of excluding eligible papers on review outcomes and (3) the criteria on which quality should be established.[68 73–75] For these reasons, the project will adopt an inclusive approach to critical appraisal, using the appraisal process to enable an in-depth understanding of each paper and to facilitate a critical, questioning approach to the study findings.[76] Studies will not be excluded on the basis of quality, rather, the quality assessment will be used: (1) to judge the relative contribution of each study to the overall synthesis and (2) to assess the methodological rigour of each study as part of a process of assessing confidence in the review findings.[77] The quality of included studies will be assessed independently by two reviewers using the Joanna Briggs Institute Qualitative Assessment and Review Instrument.[78 79] This tool has been found to be more coherent and more sensitive to assessment of validity than other commonly used tools.[75] An extract summary of the appraisal items is listed in figure 4.

## Data extraction

Data extraction will primarily be undertaken by one reviewer with a second and third reviewer each checking a random sample of the articles for completeness in extraction. Data will be extracted using a modified JBI data extraction form (figure 5) to include details of the phenomenon of interest, population, context, study methodology, methods and findings of significance to the review objectives.[78 80] If studies lack relevant detail, authors will be contacted to provide further information. The 'results/findings' sections from each study will be entered into NVivo for further analysis and synthesis. There is ongoing debate over what constitutes a 'finding' in qualitative research and how to differentiate between findings—that is, the study authors' analysis/interpretation of the primary data and other inferences or conclusions made by the authors.[72 81–83] In this review, we will follow the approach advocated by Thomas and Harden[57] in which the 'results/findings' section of each paper will be entered into NVivo for detailed thematic analysis (described below). However, if, on reading a paper, it becomes clear that a finding is reported elsewhere (eg, in the abstract or discussion section), this excerpt will also be extracted into NVivo.

Where a paper presents findings on both women's and health professional's experiences, the relevant sections will be extracted for each review separately where possible.

## Synthesis

Analysis will be conducted by two reviewers in consultation with the research team. We shall carry out an inductive-thematic analysis for each review,[57] conducted in four phases: (1) in-depth reading and immersion in the data, (2) coding of findings that are connected to FGM-related healthcare, (3) recoding to higher level themes and (4) interpretive synthesis. If appropriate, the latter stage may involve the development of models or frameworks to identify and display relationships between, and patterns within, the analytical themes. Where possible, the analytical themes will be formulated as 'directive' findings indicating clear messages and/or suggesting clear lines of action for policy and practice.[78]

## Assessment of confidence

The review findings will be assessed using the GRADE-CER-QUAL guidelines.[77] The assessment of confidence in the evidence for an individual review finding considers four elements: (1) methodological limitations (the extent to which there are problems in the design or conduct of primary studies that contributed to evidence of a review finding); (2) relevance (the extent to which the body of evidence from the primary studies supporting a review finding is applicable to the context specified in the review question); (3) coherence (whether the finding is well grounded in data from the primary studies and can provide a convincing explanation for pattern found in the data) and (4) adequacy of data (an overall determination of the degree of richness and quantity of data supporting a review finding).[77] Based on the assessment, each review finding will be assigned one of the four levels of confidence: high, moderate, low and very low.

## PRESENTING AND REPORTING THE REVIEW

Each review will be reported in accordance with the 'Enhancing Transparency in Reporting the Synthesis of Qualitative Research' (ENTREQ) statement,[84] appropriate for qualitative evidence synthesis, which consists of 21 items (see table 1).

## DISSEMINATION

The participatory collaborative approach to this qualitative evidence synthesis project will enhance the multidisciplinary interpretation of findings in order to inform stakeholders and the development of FGM-care initiatives for improved confidence and quality of service provision to women and girls who have undergone FGM.

The two qualitative systematic reviews will be published separately in open access, peer reviewed, international journals and an integrated report compiled and widely disseminated to FGM stakeholders through stakeholder engagement events, round-table discussions with different stakeholder groups and tailored materials for each of the stakeholder groups (academic, health professionals, policy makers/commissioners, FGM communities and community organisations). We shall share the review findings report in the third quarter of 2018, using pre-existing contacts with the WHO Europe FGM group network and through professional networks.

**Author affiliations**
[1]School of Health Sciences, University of Nottingham, Nottingham, UK
[2]Research and Learning Services, Faculty of Medicine and Health Sciences, University of Nottingham, UK
[3]Consultant Midwife, Nottingham University Hospital Trust, Nottingham, UK
[4]Executive Board, Mojatu Foundation, Nottingham, UK

**Acknowledgements** This systematic review is funded under Project: Improving Care for Women and Girls who have undergone Female Genital Mutilation (FGM): Qualitative Evidence Synthesis. Funded by the National Institute for Health Research (NIHR) Health Service Research and Delivery Programme (NIHR 1115/137/04).

**Contributors** CE conceptualised the project and prepared the project documents as PI in collaboration with the team members GH, JM, CC and VN. JE developed the search strategy and will carry out the searches. RT prepared the manuscript with CE and input from team members. All team members reviewed and approved this manuscript.

**Funding** The National Institute Health Research (NIHR) Health Service Research and Delivery Programme (NIHR 1115/137/04).

**Competing interests** None declared.

**Provenance and peer review** Not commissioned; externally peer reviewed.

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
