## [Reviewer comments · BMJ Open]

ARTICLE DETAILS

TITLE (PROVISIONAL)	What are the experiences of seeking, receiving and providing FGM-related healthcare? Perspectives of health professionals and women / girls who have undergone FGM; Protocol for a systematic review of qualitative evidence
AUTHORS	Evans, Catrin; Tweheyo, Ritah; McGarry, Julie; Eldridge, Jeanette; McCormick, Carol; Nkoyo, Valentine; Higginbottom, Gina

VERSION 1 – REVIEW

REVIEWER	Dr Laura Jones Institute of Applied Health Research, University of Birmingham, UK Prior to being asked to review this submission, I was aware that this project had been funded. This came to light as part of a bid that I am involved in in response to a commissioned funding call from the HTA around preferences for timing of deinfibulation.
REVIEW RETURNED	06-Jul-2017

GENERAL COMMENTS	Thank you for the opportunity to review this timely and important protocol. Overall, the protocol is well structured and easy to follow for the reader. Please see below for the comments you may wish to consider. Abstract page 3 line 11: it is unclear to the reader how you have defined (and therefore would measure) “quality FGM-related healthcare”. Abstract page 3 line 14: check for typographical error Abstract page 3 line 34/5: will you also make recommendations for future research given that the reviews are likely to identify numerous gaps in the current evidence base? Strengths and limitations page 4: all of the bullet points listed here are strengths without reflection on any short comings in the proposed reviews. In addition, the clarification statement in bullet point two (“not just maternity settings”) is likely to also be relevant for bullet point one? Introduction: there are a small number of statements that lack appropriate referencing e.g. page 5 lines 8-10 “the practice is illegal...”, page 5 lines 12-18 “In the short term...” and “In the longer term...”. Please carefully check that the introduction is appropriately referenced. Introduction: it would be really beneficial for the reader to get a
---

	sense of how big the problem of FGM is globally and then perhaps in OECD countries e.g. reporting the current UNICEF estimate of how many women and girls have experienced FGM / are living with the consequences. Introduction page 7 lines 11 onwards: you quite rightly highlight that there are a number of systematic reviews now on FGM healthcare. One of the references (ref 52 Sunday-Adeoye et al.) supporting this statement is actually an editorial and not a systematic review. This editorial was published as part of a special edition around FGM in the International Journal of Gynaecology and Obstetrics in 2017. Within this special edition there were 10 systematic reviews / three qualitative evidence syntheses. Have you made reference to these publications in the protocol (from some quick searching I couldn't find them)? Although of course not all are relevant here but they do provide perhaps the most up to date evidence and may therefore warrant a mention in the protocol with some reflection on how your work will build on these data where appropriate. Objective iv page 8 lines 6-8: might you also get a sense of why FGM is not discussed within communities (especially "receiving" country FGM-affected communities) in addition to the factors influencing open discussion with HCPs? Exclusion page 12 lines 12-19: Have you thought about how will you handle mixed data e.g. papers where HCP and women's views are reported together and cannot be meaningfully disaggregated for your individual reviews? Will you exclude case studies and individual/ personal reflections which provide conceptually rich data but do not demonstrate primary qualitative data collection. Methods: there are a small number of statements that lack appropriate referencing e.g. page 8 lines 54-57 "Where the purpose..." and page 9 lines 10-17 "Synthesis involves..." and "Thematic synthesis explicitly..." Please carefully check that the methods are appropriately referenced. Dissemination page 16: Your dissemination plans appear to be UK-centric. If your reviews are going to be across a range of OECD countries then there is potential for the results to impact FGM-related care provision in a range of settings, not just the UK. When do you anticipate that the results will be published? Perhaps you could indicate this in the protocol? General comment: Within the proposed work, there is no mention of the male voice or the importance of synthesising male perspectives on FGM. I understand that this is a different question but given that your aim is to provide comprehensive recommendations for FGM-related policy and practice, this does appear to be an omission and one that should be justified. General comment: it is unclear exactly what role PPI/ wider stakeholders will have in the review. You highlight that it's a collaborative project and that stakeholders will be involved throughout the process but this is opaque currently.
--	--

REVIEWER	Caroline Homer University of Technology Sydney, Australia I am supervising a PhD student in Australia who has undertaken a
-----------------	--

	very similar review and will be submitting this for publication soon.
REVIEW RETURNED	08-Aug-2017

GENERAL COMMENTS	Thank you for the opportunity to review this protocol. The topic is important and the review is welcomed. Many aspects of the protocol are clear and well presented. Some aspects require attention:  • The section on page 7 above the aim is confusing and hard to follow. Given this is the justification for this study it needs to be addressed. • In the aim, the fact that this is focussed on care in high income countries needs to be included. Currently the aim is very broad and includes all women affected by FGM but I think the argument in the para above is that this review focuses on high OECD countries. • The focus on high OECD countries needs to be in the objectives for both Review 1 and 2. • What kind of care is the review interested in? Is that reproductive and maternity care or health care in general? This is not clear. • The authors state that the findings will provide recommendations for health professionals. It needs to be clear that these recommendations will only be from qualitative studies which are often low level in evidence in relation to interventions. • In Figure 1 – what are FGM symptoms? • The authors state that this review will take a participatory collaborative approach but this is not clear. How will this be participatory given much of the approach is reasonably standard for a narrative review.
--

REVIEWER	Dr Laura Jones Institute of Applied Health Research, University of Birmingham, UK
-----------------	--

	I am aware that both myself and the leading/corresponding author of this paper are leading separate bids in application for an NIHR HTA commissioned call exploring preferences for the timing of deinfibulation (16/78). At the time of undertaking this review the funding body had yet to announce the outcome of the applications.
REVIEW RETURNED	01-Sep-2017

GENERAL COMMENTS	Thank you for taking the time to consider and address each of the points raised. I have a very small number of further suggested revisions for your consideration. Please note that page and line references are from the 54 page proof and relate to the tracked changed version of the revised manuscript. (1) Abstract: review methods page 32 lines 28-29 & methods data extraction page 43 lines 17-18. You state that “findings” will be extracted into NVivo software. This surprised me to some extent as “typical” (although still debated) practice for qualitative evidence syntheses is to extract both findings and discussion rather than just the findings. The rationale for this is two-fold: (1) because in some qualitative papers the findings and discussion sections are blended and (2) because there may well be further interpretation from the authors presented in the discussion section that will be important to code and include in the synthesis (this is certainly my experiences of undertaking qualitative syntheses anyway). (2) Methods data extraction page 43 lines 20-22: the additional sentence about extracting women’s and HCP experiences separately might not in reality actually be possible. This is a common issue with qualitative syntheses in that mixed data are presented where the analysis and interpretation cannot be disaggregated (e.g. even if they present primary quotes separately the overall analysis may well be combined). I think that you might need to just clarify this sentence with “where possible” or “if appropriate” at the end. (3) Dissemination page 46 lines 38-43: the last paragraph (that originally came under conclusions) doesn’t really seem to fit here. If you feel strongly that it should stay in the manuscript then I wonder if this paragraph should come first in the dissemination section with the “the 2 qualitative...” paragraph coming after this.
---

VERSION 1 – AUTHOR RESPONSE

Reviewer 1 (Caroline Homer)

No further comments

Reviewer 2 (Laura Jones)

[1] Abstract: review methods page 32 lines 28-29 & methods data extraction page 43 lines 17-18. You state that “findings” will be extracted into NVivo software. This surprised me to some extent as “typical” (although still debated) practice for qualitative evidence syntheses is to extract both findings and discussion rather than just the findings. The rationale for this is two-fold: (1) because in some qualitative papers the findings and discussion sections are blended and (2) because there may well be further interpretation from the authors presented in the discussion section that will be important to code and include in the synthesis (this is certainly my experiences of undertaking qualitative syntheses anyway).

Response: Thank you for this comment. As you rightly point out, this is an area that is under debate. Different authors aligned to different review methodologies and ontological positions (ranging from idealist to realist) advocate different approaches to this issue – as acknowledged in the Cochrane Qualitative and Implementation Methods Group Handbook (2011). Within thematic synthesis (the approach that we will follow), published reviews frequently follow the process advocated by Thomas & Harden (2009) – where the ‘findings’ sections of a paper are extracted for further analysis into NVivo (or EPPI software). Reviews adopting this approach have been published in the Cochrane Database and the BMJ, amongst other well reputed journals (see a couple of selected references below). Hence we respectfully suggest that our approach is justified and is in line with that of other research groups (even if other review teams may choose to take a different approach). However, to clarify and theoretically justify our stance, we have elaborated the relevant text in the protocol as follows: The findings from each study will then be extracted. There is on-going debate over what constitutes a ‘finding’ in qualitative research and how to differentiate between findings - i.e. the study authors’ analysis/interpretation of the primary data and other inferences or conclusions made by the authors 72 81-83. In this review, we will follow the approach advocated by Thomas & Harden 57 in which the ‘results/findings’ section of each paper will be entered into NVivo for detailed thematic analysis (described below). However, if, upon reading a paper, it becomes clear that a finding is reported elsewhere (e.g. in the abstract or discussion section) this excerpt will also be extracted into NVivo.

- Horton R, Tong A, Howard K, et al. The views of patients and carers in treatment decision making for chronic kidney disease: systematic review and thematic synthesis of qualitative studies. *BMJ* 2010;340:11
- Jordan R, Dainty K, Noyes J, et al. Factors that impact on the use of mechanical ventilation weaning protocols in critically ill adults and children: a qualitative evidence-synthesis. *Cochrane Database of Systematic Reviews* 2016(10)
- Noyes J, Booth A, Hannes K, et al. Supplementary guidance for inclusion of qualitative research in Cochrane systematic reviews of interventions, Version 1: Cochrane Collaboration Qualitative Methods Group, 2011
- Sandelowski M, Barroso J. Finding the findings in qualitative studies. *Journal of Nursing Scholarship* 2002;34:213-19.
- Thomas J, Harden A. Methods for the thematic synthesis of qualitative research in systematic reviews. *BMC Medical Research Methodology* 2009;8:45.

Reviewer 2 (Laura Jones)

Comment [2] Methods data extraction page 43 lines 20-22: the additional sentence about extracting women’s and HCP experiences separately might not in reality actually be possible. This is a common issue with qualitative syntheses in that mixed data are presented where the analysis and interpretation cannot be disaggregated (e.g. even if they present primary quotes separately the overall

analysis may well be combined). I think that you might need to just clarify this sentence with “where possible” or “if appropriate” at the end.

Response: We agree with your point and suggestion. Accordingly, we have added in the words “where possible”.

Reviewer 2 (Laura Jones)

Comment [3] Dissemination page 46 lines 38-43: the last paragraph (that originally came under conclusions) doesn’t really seem to fit here. If you feel strongly that it should stay in the manuscript then I wonder if this paragraph should come first in the dissemination section with the “the 2 qualitative...” paragraph coming after this.

Response: As suggested, we have changed the order of the paragraphs.

VERSION 2 – REVIEW

REVIEWER	Laura Jones University of Birmingham, UK As previously highlighted, the lead author and I have led competing bids for an HTA Commissioned Call exploring timing preferences for deinfibulation.
REVIEW RETURNED	16-Oct-2017
GENERAL COMMENTS	Thank you for taking the time to consider each of the comments.